# COLA: CONSISTENT LEARNING WITH OPPONENT-LEARNING AWARENESS

## ABSTRACT

Optimization problems with multiple, interdependent losses, such as Generative Adversarial Networks (GANs) or multi-agent RL, are commonly formalized as *differentiable games*. Learning with Opponent-Learning Awareness (LOLA) introduced *opponent shaping* to this setting. More specifically, LOLA introduced an augmented learning rule that accounts for the agent's influence on the anticipated learning step of the other agents. However, the original LOLA formulation is *inconsistent* because LOLA models other agents as *naive learners* rather than LOLA agents. In previous work, this inconsistency was suggested as a root cause of LOLA's failure to preserve stable fixed points (SFPs). We show that, contrary to claims in previous work, Competitive Gradient Descent (CGD) does not solve the consistency problem and does not recover high-order LOLA (HOLA) as a series expansion. Working towards a remedy, we formalize consistency and show that HOLA is consistent whenever it converges; however, it may fail to converge altogether. We propose a new method called Consistent LOLA (COLA) which *learns* update functions that are consistent under mutual opponent shaping. We prove that even such consistent update functions do not preserve SFPs, contradicting the hypothesis that this shortcoming is due to inconsistency. Finally, we empirically compare the performance and consistency of aforementioned algorithms on a range of general-sum learning games.

## 1 INTRODUCTION

Multi-objective problems can be found in many domains, such as GANs (Goodfellow et al., 2014) or single- and multi-agent reinforcement learning (RL) in the form of imaginative agents (Racanière et al., 2017), hierarchical RL (Barto & Mahadevan, 2002), and intrinsic curiosity (Schmidhuber, 1991). A popular framework to understand systems with multiple, interdependent losses is differentiable games (Balduzzi et al., 2018). For example, in the case of GANs, the differentiable game framework models the generator and the discriminator as competing agents, each trying to optimize their respective loss. The action space of the game consists of choosing the respective network parameters (Balduzzi et al., 2018).

An effective paradigm to improve learning in differentiable games is *opponent shaping*, where the players use their ability to shape each other's learning steps. LOLA (Foerster et al., 2018) was the first work to make explicit use of opponent shaping in the differentiable game setting. LOLA is also one of the only general learning methods designed for differentiable games that obtains mutual cooperation with the Tit-for-Tat strategy in the Iterated Prisoner's Dilemma (IPD). The Tit-for-Tat strategy starts out cooperating and retaliates once whenever the opponent does not cooperate. It achieves mutual cooperation and has proven to be successful at IPD tournaments (Axelrod, 1984; Harper et al., 2017). In contrast, naive gradient descent and other more sophisticated methods typically converge to the mutual defection policy under random initialization (Letcher et al., 2019b).

While LOLA discovers these interesting equilibria, the original LOLA formulation is inconsistent because LOLA agents assume that their opponent is a *naive learner*. This assumption is clearly violated if two LOLA agents learn together in a game. It has been suggested that this inconsistency is the root cause for LOLA's shortcomings, such as not converging to SFPs in some simple quadratic games (Letcher 2018, p. 2, 26; see also Letcher et al. 2019b).

**Contributions.** How can LOLA's inconsistency be resolved? To answer this question, we first revisit the concept of *higher-order* LOLA (HOLA) (Foerster et al., 2018) in Section 4.1. For example, *second-order* LOLA assumes that the opponent is a *first-order* LOLA agent (which in turn assumes the opponent is a naive learner) and so on. Assuming that HOLA converges with increasing order, we define *infinite-order* LOLA (iLOLA) as the limit of HOLA whenever it exists. Intuitively, it should follow that two iLOLA agents have a consistent view of each other, meaning they make an accurate assumption about the learning behavior of the opponent under mutual opponent shaping. We introduce a formal definition of consistency and prove that iLOLA is indeed self-consistent under mutual opponent shaping.

Previous work has claimed that a series expansion of Competitive Gradient Descent (CGD) (Schäfer & Anandkumar, 2020) recovers high-order LOLA. This would imply that CGD corresponds to iLOLA and thus solves the consistency problem. In Section 4.2, we prove that this is false: CGD does not in general correspond to iLOLA, and, unlike iLOLA, does not resolve the problem of consistency. In particular, we show that, contrary to previous claims, the series expansion of CGD does *not* correspond to higher-order LOLA.

There are a number of problems with addressing consistency using a limiting update (iLOLA): the process may not converge, and requires computation of arbitrarily high derivatives. In Section 4.3, we propose Consistent LOLA (COLA) as a more general and efficient alternative. Instead of repeatedly applying the LOLA learning rule (iLOLA), COLA learns a pair of consistent update functions by explicitly minimizing a consistency loss. By reframing the problem as such, the method only requires up to second-order derivatives, and instead of having a handcrafted update function as for LOLA or CGD, we use the representation power of neural networks to *learn* the update step.

In Section 4.4, we prove initial results about COLA. First, we show that COLA's solutions are not necessarily unique. Second, despite being consistent, COLA does not recover SFPs, contradicting the prior belief that this shortcoming is caused by inconsistency. Third, we provide an example in which COLA converges more robustly, i.e., under a wider range of learning rates, than LOLA.

Finally, in Sections 5 and 6, we report our experimental setup and results, investigating COLA and HOLA and comparing it to LOLA and CGD in a range of games. We show that, despite its non-uniqueness, COLA tends to find similar solutions in different runs empirically. Moreover, we show that COLA finds the iLOLA solution when HOLA converges but finds different solutions when HOLA diverges. These solutions have lower consistency loss and converge under a broader range of learning rates than LOLA and HOLA. Our experiments also show that, while COLA does not find Tit-for-Tat on the IPD (unlike LOLA), it does learn policies with near-optimal total payoff.

## 2 RELATED WORK

General-sum learning algorithms and their consequences have been investigated from different perspectives in the reinforcement learning, game theory, and GAN literature, see e.g. (Schmidhuber, 1991; Barto & Mahadevan, 2002; Racanière et al., 2017; Goodfellow et al., 2014) to name a few. Next, we will highlight a few of the approaches to the mutual opponent shaping problem.

Opponent modeling maintains an explicit belief of the opponent, which allows to reason over their strategies and compute optimal responses. Opponent modeling can be divided into different subcategories: There are classification methods, classifying the opponents into pre-defined types (Weber & Mateas, 2009; Synnaeve & Bessiere, 2011), or policy reconstruction methods, where we explicitly predict the actions of the opponent (Mealing & Shapiro, 2017). Most closely related to opponent shaping is recursive reasoning, where methods model nested beliefs of the opponents (He et al., 2016; Albrecht & Stone, 2019; Wen et al., 2019).

In comparison, COLA assumes that we have access to the ground-truth model of the opponent, e.g., the opponent's payoff function, parameters, and gradients, which puts COLA into the framework of differentiable games (Balduzzi et al., 2018). Various methods have been proposed, investigating the local convergence properties to different solution concepts (Mescheder et al., 2018; Mazumdar et al., 2019; Letcher et al., 2019b; Azizian et al., 2020; Schäfer & Anandkumar, 2020; Schäfer et al., 2020; Hutter, 2020). Most of the work in differentiable games has not focused on the issue of opponent shaping and consistency. Mescheder et al. (2018) and Mazumdar et al. (2019) focus solely on zero-sum games without shaping. Letcher et al. (2019b) improve on LOLA, but do not investigate the

consistency issue. CGD (Schäfer & Anandkumar, 2020) addresses the consistency issue of LOLA for zero-sum games but not for general-sum games. The exact difference between CGD and LOLA is addressed in the Section 4.2.

## 3 BACKGROUND

### 3.1 DIFFERENTIABLE GAMES

The framework of differentiable games has become increasingly popular to model the problem of multi-agent learning. Whereas in the framework of stochastic games we are typically limited to parameters such as action-state probabilities, differentiable game generalizes to any parameters as long as the loss function is differentiable with respect to them (Balduzzi et al., 2018). We restrict our attention on two-player games, as is standard in the current differentable games literature.

**Definition 1** (Differentiable games)**.** In a two-player differentiable game, players $i = 1, 2$ control parameters $\theta_i \in \mathbb{R}^{d_i}$ to minimize twice continuously differentiable losses $L^i : \mathbb{R}^{d_1 + d_2} \to \mathbb{R}$. We adopt the convention to write $-i$ to denote the respective other player.

A fundamental challenge of the multi-loss setting is finding a meaningful solution concept. Whereas in the single loss setting the typical solution concept is local minima, in multi-loss settings there are different sensible solution concepts. Most prominently, there are Nash Equilibria (Osborne & Rubinstein, 1994). However, Nash Equilibria include unstable saddle points that cannot be reasonably found via gradient-based learning algorithms (Letcher et al., 2019b). A more appropriate concept are stable fixed points (SFPs), which could be considered a differentiable game analogon to local minima in single loss optimization. We will omit a formal definition here for brevity and point the interested reader to previous work on the topic (Letcher et al., 2019a).

### 3.2 LOLA AND SOS

Consider a differentiable game with two players. A LOLA agent $\theta_1$ uses its access to the opponent's parameters $\theta_2$ to differentiate through the learning step of the opponent. In other words, agent 1 reformulates their loss to $L^1(\theta_1, \theta_2 + \Delta\theta_2)$, where $\Delta\theta_2$ represents the assumed learning step of the opponent. In first-order LOLA we assume the opponent to be a naive learner: $\Delta\theta_2 = -\alpha\nabla_2 L^2$, which is what makes LOLA inconsistent if the opponent was any other type of learner. Note that $\nabla_2$ denotes the gradient with respect to $\theta_2$. Also note that $\alpha$ represents the *look-ahead* rate, which is the assumed learning rate of the opponent. In the original paper the loss was approximated using a Taylor expansion $L^1 + (\nabla_2 L^1)^\top \Delta\theta_2$. For agent 1, their first-order (Taylor) LOLA update is then defined as

$$\Delta\theta_1 := -\alpha\left(\nabla_1 L^1 + \nabla_{12} L^1 \Delta\theta_2 + (\nabla_1 \Delta\theta_2)^\top \nabla_2 L^1\right).$$

Alternatively, in *exact* LOLA, the derivative is taken directly with respect to $L^1(\theta_1, \theta_2 + \Delta\theta_2)$.

LOLA has had some empirical success, being one of the first general learning methods to discover Tit-for-Tat like solutions in social dilemmas. However, later work showed that LOLA does not preserve SFPs $\bar{\theta}$ since the rightmost term can be nonzero at $\bar{\theta}$. In fact, LOLA agents show "arrogant" behavior: they assume they can shape the learning of their *naive* opponents without having to adapt to the shaping of the opponent. Prior work hypothesized that this arrogant behavior is due to LOLA's inconsistent formulation (Letcher 2018, p. 2, 26; see also Letcher et al. 2019b).

To improve upon LOLA, Letcher et al. (2019b) have suggested the Stable Opponent Shaping (SOS) algorithm. SOS applies a correction to the LOLA update, leading to theoretically guaranteed convergence to SFPs. However, despite its desirable convergence properties, SOS still does not solve the conceptual issue of inconsistent assumptions about the opponent.

### 3.3 CGD

CGD (Schäfer & Anandkumar, 2020) proposes updates that are themselves Nash Equilibra of a local bilinear approximation of the game. It stands out by its robustness to different step sizes of opponents and its ability to find SFPs. However, CGD does not find Tit-for-Tat on the IPD, instead

Table 1: (a) This table shows the log of the squared consistency loss on the Tandem game, where e.g. HOLA6 is sixth-order higher-LOLA. (b) Cosine similarity between COLA and LOLA, HOLA2, and HOLA6 over different look-ahead rates on the Tandem game.

| | (a) | | | | | (b) | | |
|---|---|---|---|---|---|---|---|---|
| $\alpha$ | LOLA | HOLA2 | HOLA6 | COLA | $\alpha$ | LOLA | HOLA2 | HOLA4 |
| 1.0 | 128.0 | 512 | 131072 | 4.84e-14 | 1.0 | 0.57 | 0.58 | 0.60 |
| 0.5 | 12.81 | 14.05 | 12.35 | 2.62e-14 | 0.5 | 0.61 | 0.46 | 0.15 |
| 0.3 | 2.61 | 2.05 | 0.66 | 4.09e-14 | 0.3 | 0.92 | 0.51 | 0.72 |
| 0.1 | 0.08 | 9.13e-3 | 1.62e-6 | 6.55e-14 | 0.1 | 0.94 | 0.98 | 0.99 |
| 0.01 | 1.41e-5 | 2.10e-8 | 3.69e-14 | 8.58e-14 | 0.01 | 0.99 | 1.0 | 1.0 |

converging to mutual defection (see Figure 13 in Appendix I.6). CGD's update rule can be written as

$$\begin{pmatrix} \Delta\theta_1 \\ \Delta\theta_2 \end{pmatrix} = -\alpha \begin{pmatrix} \mathrm{Id} & \alpha\nabla_{12}L^1 \\ \alpha\nabla_{21}L^2 & \mathrm{Id} \end{pmatrix}^{-1} \begin{pmatrix} \nabla_1 L^1 \\ \nabla_2 L^2 \end{pmatrix} \tag{1}$$

One can recover different orders of CGD by approximating the inverse matrix via the series expansion $\|A\| < 1 \Rightarrow (\mathrm{Id} - A)^{-1} = \lim_{N\to\infty} \sum_{k=0}^N A^k$. For example, at N=1, we recover a version called Linearized CGD (LCGD), defined via $\Delta\theta_1 := -\alpha\nabla_1 L^1 + \alpha^2\nabla_{12}L^1\nabla_2 L^2$.

## 4 METHOD AND THEORY

### 4.1 CONVERGENCE AND CONSISTENCY OF HIGHER-ORDER LOLA

To begin, we define and analyze iLOLA. In this section, we focus on exact LOLA, but we provide a version of our definition and proof of consistency for Taylor LOLA in Appendix C. HOLA$n$ is defined by the recursive relation

$$h_1^n := -\alpha\nabla_1 \left( L^1(\theta_1, \theta_2 + h_2^{n-1}) \right)$$
$$h_2^n := -\alpha\nabla_2 \left( L^2(\theta_1 + h_1^{n-1}, \theta_2) \right)$$

with $h_1^{-1} = h_2^{-1} = 0$, omitting arguments $(\theta^1, \theta^2)$ for convenience. In particular, HOLA0 coincides with simultaneous gradient descent while HOLA1 coincides with LOLA.

**Definition 2** (iLOLA). If HOLA$n = (h_1^n, h_2^n)$ converges pointwise as $n \to \infty$, define

$$\mathrm{iLOLA} := \lim_{n\to\infty} \begin{pmatrix} h_1^n \\ h_2^n \end{pmatrix} \text{ as the limiting update.}$$

We show in Appendix A that HOLA does not always converge, even in simple quadratic games. On the other hand, iLOLA satisfies a criterion of *consistency* whenever HOLA does converge (under some assumptions), formally defined as follows:

**Definition 3** (Consistency). Any update functions $h_1 \colon \mathbb{R}^d \to \mathbb{R}^{d_1}$ and $h_2 \colon \mathbb{R}^d \to \mathbb{R}^{d_2}$ are *consistent* (under mutual opponent shaping) if for all $\theta_1 \in \mathbb{R}^{d_1}, \theta_2 \in \mathbb{R}^{d_2}$, they satisfy

$$h_1 = -\alpha\nabla_1(L^1(\theta_1, \theta_2 + h_2)) \tag{2}$$
$$h_2 = -\alpha\nabla_2(L^2(\theta_1 + h_1, \theta_2)) \tag{3}$$

**Proposition 1.** *Let* HOLA$n = (h_1^n, h_2^n)$ *denote player $i$'s exact $n$-th order LOLA update. Assume that $\lim_{n\to\infty} h_i^n(\theta) = h_i(\theta)$ and $\lim_{n\to\infty} \nabla_i h_{-i}^n(\theta) = \nabla_i h_{-i}(\theta)$ exist for all $\theta$ and $i \in \{1, 2\}$. Then iLOLA is consistent under mutual opponent shaping.*

*Proof.* In Appendix B. □

### 4.2 CGD DOES NOT RECOVER HIGHER-ORDER LOLA

Schäfer & Anandkumar (2020) claim that "LCGD coincides with first order LOLA" (page 6), and moreover that the "series-expansion [of CGD] would recover higher-order LOLA" (page 4). Unfortunately, we prove that this is untrue in general games. LCGD coincides instead with LookAhead

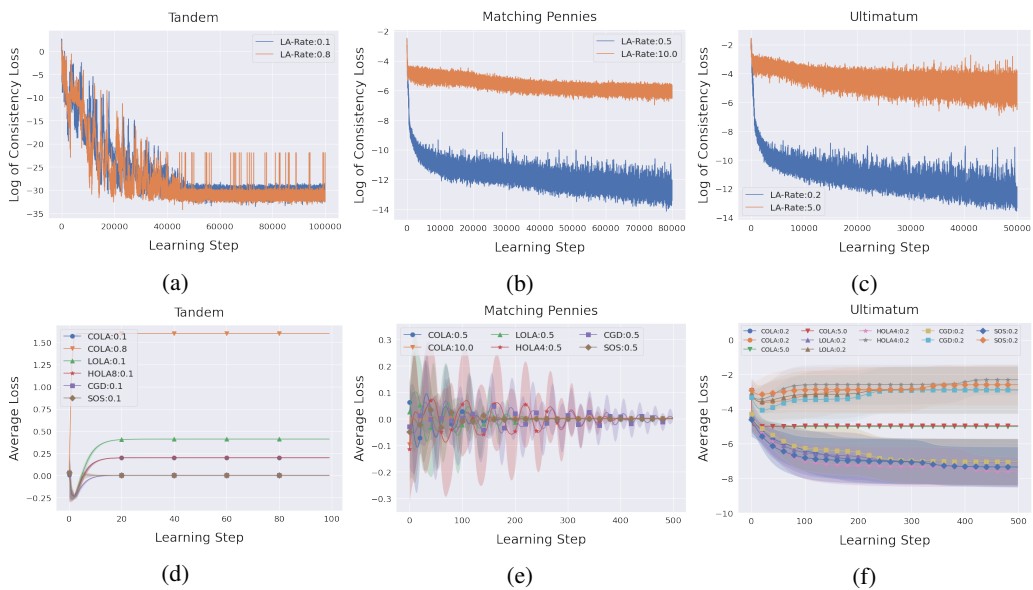

Figure 1: Subfigure (a), (b) and (c) depicts the log of the consistency loss over the training of the update functions for the Tandem, MP and Ultimatum games. Subfigure (d), (e) and (f) show the performance of COLA in comparison to HOLA:0.1, LOLA:0.1 and CGD:0.1. COLA:0.1 denotes COLA with a look-ahead rate of 0.1.

(Zhang & Lesser, 2010), an algorithm that lacks opponent shaping. Similarly, the series-expansion of CGD recovers high-order LookAhead but not high-order LOLA (neither exact nor Taylor).

**Proposition 2.** *In general, CGD is inconsistent and does not coincide with iLOLA. In particular, the series-expansion of CGD does not recover HOLA (but does recover high-order LookAhead). Moreover, LCGD does not coincide with LOLA (but does coincide with LookAhead).*

*Proof.* In Appendix D. For the negative results, it suffices to construct a single counter-example: we show that LCGD and LOLA differ almost-everywhere in the Tandem game (excluding a set of measure zero). Proving that the series-expansion of CGD does not recover HOLA relies on noticing that this would imply CGD satisfying the consistency equations for $\alpha$ sufficiently small. We prove that this also fails almost-everywhere in the Tandem game. We then show that LCGD = LookAhead and that the series-expansion of CGD recovers high-order LookAhead in general games. □

### 4.3 COLA

iLOLA is consistent under mutual opponent shaping. However, HOLA does not always converge and, even when it does, it may be expensive to recursively compute HOLA$n$ for sufficiently high $n$ to achieve convergence.

As an alternative, we propose consistent LOLA (COLA). COLA finds consistent update functions and avoids an infinite regress by directly solving the equations in Definition 3 numerically. To do so, we define the *consistency losses* for a pair of update functions $(h_1, h_2)$ parameterized by $(\phi_1, \phi_2)$, obtained for a given $\theta$ as the difference between RHS and LHS in Definition 3:

$$C_1(\phi_1, \phi_2, \theta_1, \theta_2) = \left\| h_1(\theta_1, \theta_2) + \alpha \nabla_1(L^1(\theta_1, \theta_2 + h_2(\theta_1, \theta_2))) \right\| \quad (4)$$

$$C_2(\phi_1, \phi_2, \theta_1, \theta_2) = \left\| h_2(\theta_1, \theta_2) + \alpha \nabla_2(L^2(\theta_1 + h_1(\theta_1, \theta_2), \theta_2)) \right\|. \quad (5)$$

If both losses are minimised to 0 for all $\theta$, then the two update functions are consistent. For this paper, we define $h_1$ and $h_2$ as neural networks parameterized by $\phi_1$ and $\phi_2$ respectively, and numerically minimize the sum of both losses over a region of interest using Adam (Kingma & Ba, 2017).

Table 2: (a) Comparison of consistency losses over multiple look-ahead rates on the MP game. (b) Cosine similarity between COLA and LOLA, HOLA2 and HOLA4 over different look-ahead rates on the MP game.

<table>
<tr><td colspan="5" align="center">(a)</td><td colspan="4" align="center">(b)</td></tr>
<tr><td>$\alpha$</td><td>LOLA</td><td>HOLA2</td><td>HOLA4</td><td>COLA</td><td>$\alpha$</td><td>LOLA</td><td>HOLA2</td><td>HOLA4</td></tr>
<tr><td>10</td><td>0.06</td><td>0.70</td><td>6.56</td><td>0.24</td><td>10</td><td>0.90</td><td>0.87</td><td>0.68</td></tr>
<tr><td>5</td><td>4.59e-3</td><td>0.03</td><td>0.15</td><td>9.47e-3</td><td>5</td><td>0.98</td><td>0.95</td><td>0.89</td></tr>
<tr><td>1.0</td><td>8.79e-6</td><td>3.25e-8</td><td>4.37e-9</td><td>2.35e-7</td><td>1.0</td><td>0.99</td><td>0.99</td><td>0.99</td></tr>
<tr><td>0.5</td><td>4.80e-7</td><td>2.53e-10</td><td>5.18e-12</td><td>1.30e-7</td><td>0.5</td><td>0.99</td><td>0.99</td><td>0.99</td></tr>
<tr><td>0.01</td><td>1.07e-13</td><td>5.58e-17</td><td>5.30e-17</td><td>6.99e-8</td><td>0.01</td><td>0.99</td><td>0.99</td><td>0.99</td></tr>
</table>

The parameter region of interest $\Theta$ depends on the game being played. For a game with probabilities as actions, we select an area that captures most of the probability space (e.g. we sample a pair of parameters $(\theta_1, \theta_2) \sim [-7, 7]$ as $\sigma(7) \approx 1$, where $\sigma$ is the Sigmoid function).

The expected aggregate consistency loss over the region is then defined as

$$C(\phi_1, \phi_2) = \mathbb{E}_{(\theta_1, \theta_2) \sim \mathcal{U}(\Theta)} \left[ C_1(\phi_1, \phi_2, \theta_1, \theta_2) + C_2(\phi_1, \phi_2, \theta_1, \theta_2) \right]. \qquad (6)$$

We optimize this loss by sampling parameter pairs $(\theta_1, \theta_2)$ uniformly from $\Theta$ and feeding them to the neural networks $h_1$ and $h_2$, each outputting the parameter update for an agent. We then update $\phi_1, \phi_2$ by taking a gradient step to minimize $C$.

We train the update functions until the loss has converged. We then use the learned update functions to train a pair of agent policies in the given game.

## 4.4 THEORETICAL RESULTS ABOUT COLA

In this section, we provide theoretical results about COLA's uniqueness and convergence behavior, using the Tandem game (Letcher et al., 2019b) and the Hamiltonian game (Balduzzi et al., 2018) as examples. These are simple 1-dimensional quadratic resp. bilinear games, with losses given in Section 5. Proofs for the following propositions can be found in Appendices E, F and G, respectively.

First, we show that solutions to the consistency equations are in general not unique, even when restricting to linear update functions in the Tandem game. Interestingly, empirically, COLA does seem to consistently converge to the same solution regardless (see Table 7 in Appendix I.3).

**Proposition 3.** *Solutions to the consistency equations are not unique, even when restricted to linear solutions; more precisely, there exist several linear consistent solutions to the Tandem game.*

Second, we show that consistent solutions do not always preserve SFPs, contradicting the hypothesis that LOLA's failure to preserve SFPs is due it its inconsistency (see Section 3.2). We support this result experimentally in Section 6.

**Proposition 4.** *Consistency does not imply preservation of stable fixed points: there is a consistent solution to the Tandem game with $\alpha = 1$ that fails to preserve **any** SFP. Moreover, for any $\alpha > 0$, there are **no** linear consistent solutions to the Tandem game that preserve more than one SFP.*

Third, we show that COLA can have more robust convergence behavior than LOLA and SOS:

**Proposition 5.** *For any non-zero initial parameters and any $\alpha > 1$, LOLA and SOS have divergent iterates in the Hamiltonian game. By contrast, any linear solution to the consistency equations converges to the origin for any initial parameters and **any** look-ahead rate $\alpha > 0$; moreover, the speed of convergence strictly increases with $\alpha$.*

## 5 EXPERIMENTS

We carry out our investigation on a set of games from the literature (Balduzzi et al., 2018; Letcher et al., 2019b) using SOS and CGD as baselines. For details on the training procedure of COLA, we refer the reader to Appendix H.

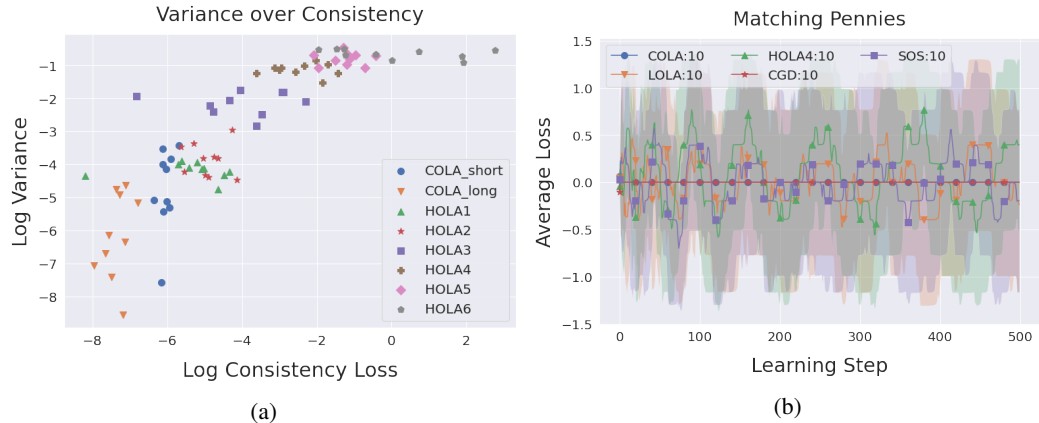

Figure 2: Training in MP at look-ahead rate of $\alpha = 10$. (a) Axes are on a log-scale. Increasing the consistency helps with decreasing the variance of the solution. (b) LOLA and HOLA find non-convergent or even diverging solutions, while COLA is convergent.

First, we compare HOLA and COLA on quadratic, general-sum games, including the Tandem game (Letcher et al., 2019b), where LOLA fails to converge to SFPs. Second, we investigate non-quadratic games, such as the zero-sum Matching Pennies (MP) game, the general-sum Ultimatum game (Hutter, 2020) and the iterated prisoner's dilemma (IPD) (Axelrod, 1984; Harper et al., 2017).

We investigate the convergence behavior of HOLA and COLA by comparing the consistency losses over a range of look-ahead rates, where COLA is retrained for each look-ahead rate to ensure a fair comparison. To compare the solutions found by HOLA and COLA, we compute the cosine similarity between the two across randomly sampled parameters across our region of interest.

**Quadratic and bilinear games.** Losses in the **Tandem game** (Letcher et al., 2019b) are given by

$$L^1(x, y) = (x + y)^2 - 2x \quad \text{and} \quad L^2(x, y) = (x + y)^2 - 2y \tag{7}$$

for agent 1 and 2 respectively. The Tandem game was originally introduced to show that LOLA fails to preserve SFPs at $x + y = 1$ and instead converges to sub-optimal solutions (Letcher et al., 2019b). Additionally to the Tandem game, we investigate the algorithms on the **Hamiltonian game**, $L^1(x, y) = xy$ and $L^2(x, y) = -xy$; and the **Balduzzi game**, where $L^1(x, y) = \frac{1}{2}x^2 + 10xy$ and $L^2(x, y) = \frac{1}{2}y^2 - 10xy$ (Balduzzi et al., 2018).

**Matching Pennies.** The payoff matrix for the Matching Pennies (MP) (Lee & K, 1967) game is shown in Appendix I.3 in Table 6. Each policy is parameterized with a single parameter, the log-odds of choosing heads $p_{\text{heads}} = \sigma(\theta_A)$. In this game, the unique Nash equilibrium is playing heads half the time.

**Ultimatum game.** The binary, single-shot Ultimatum game (Güth et al., 1982; Sanfey et al., 2003; Oosterbeek et al., 2004; Henrich et al., 2006) is set up as follows. There are two players, player A and B. Player A has access to \$10. They can split the money fairly with B (\$5 for each player) or they can split it unfairly (\$8 for player A, \$2 for player B). Player B can either accept or reject the proposed split. If player B rejects, the reward is 0 for both players. If player B accepts, the reward follows the proposed split. Player A's parameter is the log-odds of proposing a fair split $p_{\text{fair}} = \sigma(\theta_A)$. Player B's parameter is the log-odds of accepting the unfair split (assuming that player B always accepts fair splits) $p_{\text{accept}} = \sigma(\theta_B)$.

$$V_A = 5p_{\text{fair}} + 8(1 - p_{\text{fair}})p_{\text{accept}} \quad \text{and} \quad V_B = 5p_{\text{fair}} + 2(1 - p_{\text{fair}})p_{\text{accept}} \tag{8}$$

**IPD.** We next investigate the infinitely iterated prisoners' dilemma with discount factor $\gamma = 0.96$ and the usual payout function (see Appendix I.6). An agent $i$ is defined through 5 parameters, the log-odds of cooperating for the first time step and for the four possible tuples of past actions of both players in the later steps.

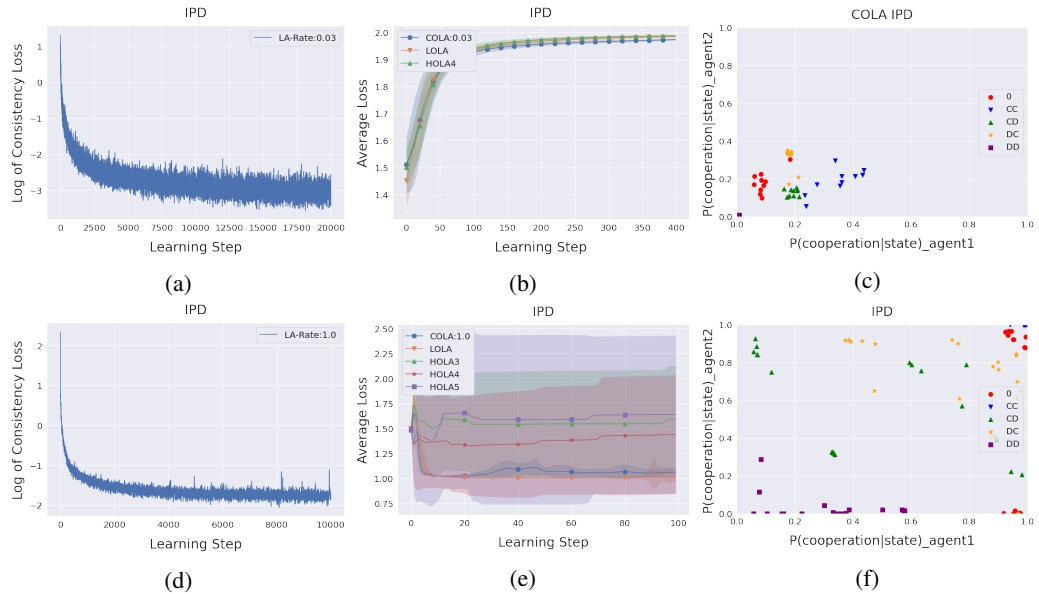

Figure 3: Results are on the IPD. Subfigure (a) / (d), show the consistency loss for look-ahead rate of 0.03 / 1.0 respectively, (b) / (e) the average loss and (c) / (f) the policy for the first player, both for the same pair of look-ahead rates. At low look-ahead HOLA defects and at high ones it diverges, also leading to high loss.

## 6 RESULTS

Here, we outline our experimental results, providing evidence on the empirical behavior of COLA and comparing COLA to HOLA and our baselines. Additional results can be found in Appendix I.

First, we investigate how increasing the order of HOLA affects the consistency of its updates. As we can see in Table 1a, 2a and 3a, HOLA's updates become more consistent with increasing order below a certain look-ahead rate threshold. Above that threshold, HOLA's updates become less consistent with increasing order. The threshold is game-specific. In the Tandem game, it is around a look-ahead rate of 0.5, whereas for the MP it is around 5. Such a threshold can be found empirically for all games that we evaluate on, as we show in Appendix I in Tables 4a, 5, 8a and 10. In the same Tables we observe that COLA finds consistent updates below the look-ahead threshold, though the consistency losses are higher than HOLA's for non-quadratic games. Overall the consistency losses are low enough to constitute a consistent solution. For the IPD, COLA's consistency losses are high compared to other games, but much lower than HOLA's consistency losses at high look-ahead rates. We leave it to future work to find methods that obtain more consistent solutions on the IPD. In general, COLA finds consistent updates above the look-ahead threshold even when HOLA does not.

Second, we are interested whether COLA and HOLA find similar solutions when HOLA's updates converge with increasing order. As we can see in Table 1b, 2b and 3b, they find very similar solutions measured by the cosine similarity of the respective updates over $\Theta$. Above the threshold, COLA's and HOLA's updates become less similar with increasing order of HOLA, indicating that they do not find the same solution.

Third, we analyze the solutions found by COLA qualitatively and compare to those found by LOLA, HOLA, SOS and CGD. In the Tandem game (Figure 1d), we can see that COLA finds the same solution as HOLA8, qualitatively confirming our observation from Table 1b that they find similar solutions. Moreover, COLA does not recover SFPs, thus experimentally confirming Proposition 4. COLA finds a convergent solution even at a high look-ahead rate (see COLA:0.8), whereas LOLA, HOLA and SOS do (Figure 4b in Appendix I.1). CGD is the only other algorithm in the comparison that also shows robustness to high look-ahead rates in the Tandem game.

Table 3: (a) Comparison of consistency losses over multiple look-ahead rates on the IPD game. (b) Cosine similarity between COLA and LOLA, HOLA2 and HOLA4 over different look-ahead rates, $\alpha$ on the IPD game.

| | (a) | | | | | (b) | | |
|---|---|---|---|---|---|---|---|---|
| $\alpha$ | LOLA | HOLA2 | HOLA4 | COLA | $\alpha$ | LOLA | HOLA2 | HOLA4 |
| 1.0 | 39.56 | 21.16 | 381.21 | 0.65 | 1.0 | 0.77 | 0.70 | 0.53 |
| 0.03 | 1.72e-3 | 4.72e-6 | 9.72e-8 | 0.33 | 0.03 | 0.96 | 0.98 | 0.98 |

On Matching Pennies at high look-ahead rates, SOS and LOLA mostly don't converge whereas COLA converges even faster with a high look-ahead rate (see Figure 1e and 2a).

For the Ultimatum game, the qualitative comparison shows that COLA is the only method that finds the fair solution consistently at a high look ahead rate, whereas SOS and LOLA do not (see Figure 10d in Appendix I.4). At low look-ahead rates, all algorithms find the unfair solution (Figure 1f).

For further comparison, we introduce the Chicken game in Appendix I.5. Both Taylor LOLA and SOS crash, whereas COLA swerves at high look-ahead rates (Figure 12d). Crashing in the Chicken game results in a catastrophic payout for both agents, whereas swerving results in a jointly preferable outcome.

On the IPD, all algorithms find the Defect-Defect strategy on low look-ahead rates (Figure 3b). At high look-ahead rates, COLA finds a strategy qualitatively similar to Tit-for-Tat, as displayed in Figure 3f, though much more noisy. However, COLA still achieves close to the optimal joint loss, in comparison to CGD, which finds Defect-Defect even at a high look-ahead rate (see Figure 13 in Appendix I.6).

To further motivate the point that increased consistency helps with robustness to a wider range of look-ahead rates, we plot the variance over the consistency on the Matching Pennies game at a high look-ahead rate. The variance is calculated over the trajectory of payoffs for an algorithm. The lower the consistency loss, the lower the variance of the solution. This further underlines, at least empirically, the benefits of increased consistency.

Lastly, we find empirical evidence for Proposition 5 in the Hamiltonian game (Figure 7b in Appendix I.2), where COLA converges faster at a higher look-ahead rate. Such behavior can also be seen for the Balduzzi game (Figure 6b in Appendix I.2) and the MP game (Figure 1e).

To conclude, COLA finds consistent and convergent updates over a wider range of look-ahead rates than state-of-the-art general-sum learning algorithms, such as LOLA and SOS. Furthermore, it finds qualitatively different solutions, sometimes with higher average rewards, like on the Ultimatum, Chicken and IPD games.

## 7 CONCLUSION AND FUTURE WORK

In this paper we cleared up the relation between the CGD and LOLA algorithms. We also showed that iLOLA solves part of the consistency problem of LOLA. We introduced a new method, called Consistent LOLA, that finds consistent solutions without requiring many recursive computations like iLOLA. It was believed that inconsistency leads to arrogant behaviour and lack of preservation for SFPs. We show that even with consistency, opponent shaping behaves arrogantly, pointing towards a fundamental open problem for the method.

We empirically investigated the consistency behavior of higher-order LOLA and COLA and found that HOLA's updates do not converge with increasing order in each hyperparameter regime, even for low-dimensional games with polynomial losses. Moreover, we showed that consistency increases robustness to different look-ahead rates.

This work opens more questions for future work than it answers. Some fundamental questions are the existence of solutions to the COLA equations in general games and general properties of convergence and learning outcomes. Moreover, additional work is needed to scale COLA to large settings such as GANs or Deep RL, or settings with more than two-players. Another interesting axis is addressing further inconsistent aspects of LOLA as identified in Letcher et al. (2019b).

