# OpenReview forum: "COLA: Consistent Learning with Opponent-Learning Awareness"
_ICLR.cc/2022/Conference — ICLR 2022 Submitted_

### Official Review · Reviewer_ZQJ8 · 2021-11-01

**Correctness:** 1
**Technical Novelty And Significance:** 2
**Empirical Novelty And Significance:** 2
**Recommendation:** 3
**Confidence:** 2

**Main Review:**

The paper focuses on an interesting challenge in learning in the presence of opponents.

On the other hand, the contributions the paper makes do not meet the claims made throughout the paper. For example, take section 4.1: "This is not a rigorous proof, but it should be intuitively clear..." First of all, "should" has no meaning? What if it is not? Second, this is one of the main claims that the paper is built on. If you are not going to give a rigorous proof for this, what will you give a proof for? Too much is left to future work.

The paper is also filled with similar vague language. In addition to the wishy-washy language, central concepts like "consistency" and "stability" are used throughout the paper and for the most part without a clear definition (at least for a long time). And, even comparative form, e.g., "more stable" are used even though nothing had been made quantitative.

While this shortcoming may be inherited from the literature, the assumptions (e.g., knowledge of the opponent's payoff functions, parameters, and gradients) made in the paper are just way too strict. Coupled with the lack of rigorous results, the insights established in the paper are at best limited.

There is a proposition that gives the means to check whether a pair of update functions are consistent. On the other hand, it is not clear how one actually checks the condition for consistency given in the proposition.

Some of the questions posed at the beginning of the results section are odd. It is not clear whether an empirical analysis can even ever answer such questions. As stated, only rigorous proofs would answer the questions (and the paper lacks those). Similarly, the discussion of the observations from the empirical results overgeneralizes.



**Summary Of The Paper:**

The paper is on learning in a differential game setting by accounting for the ability of the opponent to learn. It displays several issues in previously existing methods, proposes a new methods, and demonstrates some of the features that appear to be superior to the existing methods.

**Summary Of The Review:**

The paper is on an interesting issue yet it is at best a starting point and far from prime time.

----------------

After the author response:

Thanks to the authors for the response. It unfortunately does not change my assessment.

---

> ### Author Response · Authors · 2021-11-15
> **Response to Reviewer ZQJ8**
>
> Dear Reviewer ZQJ8
>
> Thank you for your valuable feedback. Next, we will address your concerns. The highlighted texts are taken from your initial review.
>
> > For example, take section 4.1: "This is not a rigorous proof, but it should be intuitively clear..." First of all, "should" has no meaning? What if if it is not?
>
> We apologize for the language used here. Section 4.1 was rewritten. Furthermore, we made the proof technically rigorous (see Appendix D, Proof of Proposition 2). For further discussion, see the joint response above. We repeat here for simplicity.
>
> “We clarified the claim that the CGD paper makes (see “Contributions” in Section 1 and Section 4.2 in general). We also added a technically rigorous proof (see Proposition 2 and  Appendix D Proof of Proposition 2) to the paper.  The proof shows that exact CGD does not in general correspond to infinite-order LOLA (iLOLA). In particular, we show that the series expansion of CGD does not recover higher-order LOLA (HOLA), contrary to claims made in the CGD paper.  Our proof applies to both Taylor HOLA and exact HOLA. “
>
> >[...] central concepts like "consistency" and "stability" are used throughout the paper and for the most part without a clear definition (at least for a long time).
>
> Thanks for the feedback. We have moved the definition of consistency to the start of our Method sections (see Section 4.1, Definition 3). The use of the word “stable” is indeed imprecise. We have replaced these mentions of stability with more appropriate descriptions depending on the context.
>
> > While this shortcoming may be inherited from the literature, the assumptions (e.g., knowledge of the opponent's payoff functions, parameters, and gradients) made in the paper are just way too strict. Coupled with the lack of rigorous result, the insights established in the paper are at best limited.
>
> This depends entirely on the setting. As the reviewer points out, it is very common to assume centralized training in the literature whereby all agents are trained in simulation and thus have access to the parameters, etc. of other agents during training.
> This is standard in the differentiable games literature, including numerous publications at top conferences (including ICLR).
>
> > There is a proposition which gives a means to check whether a pair of update functions are consistent. On the other hand, it is not clear how one actually check the condition for consistency given in the proposition.
>
> We mathematically define consistency in Definition 1 (now Definition 3, see Section 4.1). As outlined in Section 4.3, we evaluate the consistency of a given pair of update functions numerically by sampling parameters uniformly at random from a given parameter region and then evaluating the difference between the LHS and RHS of the consistency equations (see Equation 9 (now Equation 6)),
>
> > As stated, only rigorous proofs would answer the questions (and the paper lacks those). Similarly, the discussion of the observations from the empirical results overgeneralizes.
>
> We have changed the language to clarify that we are only answering the questions empirically and only on the set of games that we evaluate on. Moreover, we have added an additional proof (Proposition 3) that supports our empirical observation that COLA is empirically more robust to a wider range of look-ahead rates, at least on the Hamiltonian game, described in more detail in the joint response.

---

> ### Author Response · Authors · 2021-11-24
> **Response 2**
>
> Dear Reviewer
>
> We hope we addressed your concerns by providing multiple rigorous proofs which correct the claim made by the CGD paper and motivate our empirical observations in the experiments. We also rewrote the Results section to address your concern about overgeneralizing statements.
>
> We ask you kindly to reply to our responses as soon as possible so that we still have enough time to discuss follow-up questions. The discussion period ends on the 29th of November.
>
> Thank you so much for your time.

---

> ### Author Response · Authors · 2021-11-29
> **Response 3**
>
> Dear Reviewer
>
> The discussion period ends today on the 29th of November. We would greatly appreciate if you could consider our latest responses for a final review of our paper.
>
> Thank you for your time!

---

### Official Review · Reviewer_QYEL · 2021-11-01

**Correctness:** 3
**Technical Novelty And Significance:** 3
**Empirical Novelty And Significance:** 3
**Recommendation:** 6
**Confidence:** 3

**Main Review:**

Overall, I like the paper. However the main issue for me was that I could not clearly see the relation between CGD and LOLA. This seems to be an important correction to existing claims. However, the paper says that this is not a rigorous proof of the correction. The problem is that if an existing claims is incorrect than it is useful to correct it as soon as possible. However, the proof of correction should be quite clear. I did not find that to be the case. Ideally, it would be nice to first say that the existing claim says that series expansion will look like this and while doing so they forgot about this particular term that seems to appear in LOLA’s loss and thus the correction to the existing claim.
Overall, I think even just this correction could be a sufficient contribution for publication.

The paper claims that COLA follows a consistent update rule. However, empirically COLA does not converge to a a Tit-for-Tat strategy as desired (or as LOLA does). Then what is the motivation behind using COLA over LOLA?

I am not sure of what is the original claim form the CGD paper and if it is correctly interpreted in this paper. It seems that the original claims from the CGD paper is series expansion of CGD recovers HOLA (high order LOLA). The paper says that this implies CGD is equal to iLOLA. I am not sure how this is true.

While I have other minor points, but I think the claim regarding the original claim from CGD paper and its correction should be first made clear.

**Summary Of The Paper:**

This paper deals with the problem of learning in differentiable games. Mainly the paper tackles the problem of learning in games while taking into account the learning of the opponent as well. The main contribution of the paper is to point out a flaw in the existing claims regarding correspondence between competitive gradient descent and iLOLA. The paper further gives a definition of consistent update rules for differentiable games and based on this definition show that iLOLA is update rule is consistent. The paper proposes a new algorithm COLA and shows that it find more consistent solutions empirically.


**Summary Of The Review:**

I think my decision mainly depends on whether the claim that the paper makes a correction to the existing literature is true or not. And if the paper can it more clear.

---

> ### Author Response · Authors · 2021-11-15
> **Response to Reviewer QYEL**
>
> Dear Reviewer QYEL
>
> We appreciate your valuable feedback. Next, we will address your concerns one-by-one. The highlighted texts are taken from your feedback.
>
> > I could not clearly see the relation between CGD and LOLA.
>
> We are sorry for not having made the relationship between CGD and LOLA clearer. We further clarified the relation between CGD, its series expansion, iLOLA and higher-order LOLA in Section 4.1 along with a technically rigorous proof for our claims (see Appendix D Proof of Proposition 2).
>
> > The paper claims that COLA follows a consistent update rule. However, empirically COLA does not converge to a Tit-for-Tat strategy as desired (or as LOLA does)
>
> Recovering Tit-for-Tat is not the primary goal. There are many different strategies in the IPD, such as Grim Trigger, Win-Stay Lose-Shift, ZD-Extortion etc. The investigation into IPD is more explorative as we are interested in the consequences of increased consistency in the opponent-shaping paradigm.
>
> > Then what is the motivation behind using COLA over LOLA?
>
> First, COLA improves on LOLA conceptually as we do not assume the other agent to be a naive learner, even when he is not (e.g. two LOLA agents playing with each other). The increased consistency makes COLA more robust to a wider range of look-ahead rates as we show empirically in Table 1-3 and Table 5. Moreover, we see that COLA finds a “less arrogant” (and jointly better) solution in the Tandem game as can be seen in Figure 1d. For further details, please see the joint response above.
>
> > It seems that the original claims from the CGD paper is series expansion of CGD recovers HOLA (high order LOLA). The paper says that this implies CGD is equal to iLOLA. I am not sure how this is true.
>
> It is true that the CGD paper claims to correspond to high-order LOLA. We rewrote Section 4.1 and explicitly quoted the specific claims from the CGD paper to clarify our correction and eliminate all misunderstandings. In the newly provided proof (see Appendix D Proof of Proposition 2), we show specifically that the series expansion of CGD does not correspond to high-order LOLA and the series expansion in the limit does not correspond to iLOLA. We show that this is the case for both higher-order Taylor LOLA and higher-order exact LOLA. For further details we refer you to the joint response above. We repeat here for simplicity
>
> “We clarified the claim that the CGD paper makes (see “Contributions” in Section 1 and Section 4.2 in general). We also added a technically rigorous proof (see Proposition 2 and  Appendix D Proof of Proposition 2) to the paper.  The proof shows that exact CGD does not in general correspond to infinite-order LOLA (iLOLA). In particular, we show that the series expansion of CGD does not recover higher-order LOLA (HOLA), contrary to claims made in the CGD paper.  Our proof applies to both Taylor HOLA and exact HOLA.”

---

> ### Author Response · Authors · 2021-11-24
> **Response 2**
>
> Dear Reviewer
>
> We hope we addressed your concerns by clarifying the relations between CGD and HOLA and the claim that the CGD paper makes. Furthermore, we now provide a rigorous proof to correct the aforementioned claim. We ask you kindly to reply to our responses as soon as possible so that we still have enough time to discuss follow-up questions. The discussion period ends on the 29th of November.
>
> Thank you so much for your time.

---

### Official Review · Reviewer_bkzQ · 2021-11-02

**Correctness:** 4
**Technical Novelty And Significance:** 3
**Empirical Novelty And Significance:** 3
**Recommendation:** 8
**Confidence:** 3

**Main Review:**

### Strengths
- Paper is well written and provides a well motivated investigation into LOLA’s failure to preserve SFPs and corresponding ‘arrogant’ behavior.
- Explanation of cases where CGD is not equivalent to iLOLA in general-sum games seems important and significant.
- The motivation behind the proposed COLA method is well explained and justified, and the empirical results are thorough and support the authors’ claims.

### Weaknesses
- More thorough proof for CGD argument would strengthen the paper, as well as including it as a baseline in more of the empirical evaluations.

**Summary Of The Paper:**

The authors tackle the consistency problem of the original LOLA formulation. The paper investigates HOLA convergence, demonstrates that CGD does not correspond to high-order LOLA in general, and proposes COLA to directly address the consistency problem. The proposed method COLA seems more robust to different look-ahead values where HOLA diverges. The authors also find that COLA is still sometime susceptible to the ‘arrogant’ LOLA behavior, opening questions for future work.

**Summary Of The Review:**

This work attempts to overcome and investigate weaknesses of LOLA to tackle the consistency problem. Even with explicit consistency loss the authors still find that we still find that the arrogant behavior remains, so the insights from this investigation seem relevant for future work and open questions in this area.

---

> ### Author Response · Authors · 2021-11-15
> **Response to Reviewer bkzQ**
>
> Dear Reviewer bkzQ
>
> Thank you for the valuable and positive feedback.
>
> > More thorough proof for CGD argument would strengthen the paper, as well as including it as a baseline in more of the empirical evaluations.
>
> To address your concern we updated the paper with a rigorous proof (see Appendix D Proof of Proposition 2) to show that the series expansion of CGD does not correspond to high-order LOLA and exact CGD does not correspond to iLOLA. Please refer to the joint response for more details. Furthermore, we included CGD in the empirical evaluations for Matching Pennies, the Ultimatum game, the Chicken game and IPD (newly added, see Figure 1, Figure 7, Figure 10, Figure 12 and Figure 13).

---

### Official Review · Reviewer_hj7q · 2021-11-03

**Correctness:** 2
**Technical Novelty And Significance:** 2
**Empirical Novelty And Significance:** 2
**Recommendation:** 3
**Confidence:** 3

**Main Review:**

**Strengths:**
1. This paper studies closely related literature (i.e., LOLA, HOLA, CGD) and make interesting empirical observations to the community.
2. COLA only requires up to second-order derivatives compared to iLOLA, which requires many higher-order derivatives.

**Weaknesses/Concerns:**
1. My main concern is COLA's benefit against SOS (Letcher et al., 2019). SOS not only has a theoretical convergence guarantee to SFPs while retaining the LOLA's opponent-shaping to achieve higher performance, if needed. However, COLA does not have a convergence guarantee and, COLA may perform worse than SOS when it does not converge to SFPs (e.g., COLA:0.1 in the Tandem game). Hence, it is unclear when COLA should be used instead of SOS.
2. Could you clarify further on the consistency (Definition 1) regarding how it avoids the infinite regress problem? Specifically, Equation 5 is dependent on Equation 6 as Equation 5 includes $h_2$. Similarly, Equation 6 is dependent on Equation 5 as Equation 6 includes $h_1$. As a result, the infinite regress problem can arise when we replace $h_2$ in Equation 5 with Equation 6: to compute $h_1$ on the left-hand side of Equation 5, it requires $h_2$ in Equation 6, which then requires computation of $h_1$ in Equation 5 (and this recursion continues).
3. While it is an interesting idea to learn the update functions $h_1$ and $h_2$ via neural networks (parameterized by $\phi_1$ and $\phi_2$), I am concerned about the scalability of this approach. When the dimension of $\theta_1$ and $\theta_2$ are small, then learning the update functions that output the dimension of $\theta_1$ and $\theta_2$ is possible. However, when $\theta_1$ and $\theta_2$ are policies represented by neural networks, then the dimension of $\theta_1$ and $\theta_2$ are large, which results in the difficulty of learning $h_1$ and $h_2$.
4. In Figure 1, CGD is only compared in the Tandem domain. Why is CGD not compared in other domains, including matching pennies (competitive game)?
5. This is minor, but it is difficult to understand the experimental results because Tables 1-2 and Figure 1 are not positioned near Section 6.

**Summary Of The Paper:**

This paper investigates the inconsistency problem in LOLA: each LOLA agent assumes the other agent as a naive learner, resulting in LOLA agents not converging to SFPs in some games. This paper aims to address this problem by the infinite-order LOLA, which can have a consistent view of each other and empirically observes that HOLA may not resolve LOLA's convergence issues. Instead of HOLA, this paper proposes COLA that employs neural networks to explicitly minimize the consistency loss. COLA empirically shows that COLA finds the consistent solution when HOLA converges and finds more stable solutions when HOLA diverges.

**Summary Of The Review:**

I initially vote for a score of 3. While this paper finds interesting empirical observations to related works, I am unsure how it improves over the state-of-the-art approach in the literature, such as SOS. After reading the authors' responses to my questions, I am open to raising my score.

---

> ### Author Response · Authors · 2021-11-15
> **Response to Reviewer hj7q**
>
> Dear Reviewer hj7q
>
> Thank you for the valuable feedback. Followingly, we will highlight your concerns and address them one-by-one.
>
> > Hence, it is unclear when COLA should be used instead of SOS.
>
> Please see the joint response above, which we hope addresses your question completely.
>
> We repeat here for simplicity.
>
> “A second major concern was the advantage of COLA over SOS/LOLA. First of all, COLA is conceptually preferable since LOLA and SOS are inconsistent in their assumption that the opponent is a naive learner.
>
> Another significant benefit is COLA’s empirical robustness to a wider range of look-ahead rates. For example, in the Tandem game SOS and LOLA start diverging at a look-ahead rate of 0.5, whereas COLA still converges at look-ahead rates of 1 and higher (see Figure 4b). There is an inherent trade-off between too low look-ahead rates (no opponent shaping) and too high look-ahead rates (divergence for SOS/LOLA). Therefore robustness to higher look-ahead rates is attractive because we can introduce more opponent shaping without risking divergence. We also added a proof (see Appendix 3 Proof of Proposition 3) showing that, in contrast to LOLA and SOS, any linear solution to the consistency equations converges to the stable fixed point at the origin for any initial parameters and any look-ahead rate > 0 on the Hamiltonian game, further motivating the claim of robustness to look-ahead rates.
>
> To further empirically compare COLA, SOS and, LOLA and motivate our claim to robustness, we also extended our results. We added a figure showing that SOS and LOLA diverge on the Tandem game for a look-ahead rate of 1.0 whereas COLA converges (see Figure 4b). On Matching Pennies at high look-ahead rates, SOS and LOLA mostly don’t converge whereas COLA converges even faster with a high look-ahead rate (see Figure 2b). For the Ultimatum game we added empirical results that show that COLA is the only method that finds the fair solution consistently at a high look ahead rate, whereas SOS and LOLA do not find it (see Figure 10d). In the Chicken game (newly added), we observe that SOS and Taylor LOLA crash whereas COLA swerves at high look-ahead rates (see Figure 12d). Crashing in the Chicken game results in a catastrophic payout for both agents and is generally to be avoided, whereas swerving results in a jointly preferable outcome.”
>
> > Could you clarify further on the consistency (Definition 1) regarding how it avoids the infinite regress problem?
>
> This is a great observation! We have added a sentence to the Section on COLA to clarify how COLA avoids the infinite regress problem in solving the consistency equations (see Section 4.3 COLA).
>
> First, in Definition 1 (now Definition 3 in the updated version), we are not assigning $h_1$ any definition. We are defining what it means for two given update functions to be consistent. We can evaluate the terms on either side of the equation to determine whether the two update functions are or are not consistent, without any infinite regress. The problem you are mentioning would arise if we were to define $h_1$ and $h_2$ with “:=” in the equations.
>
> Second, you are right that one cannot solve this equation by substituting one equation into the other, as we would never be able to manipulate the equations to solve for $h_1$ and $h_2$ (similar to other differential equations). We solve this problem and avoid an infinite regress in our COLA method by turning it into an optimization problem to find a fixed point. We optimize a neural-network function $h_1,h_2$ by performing gradient descent on the difference between the LHR and RHS of the consistency equations. During each optimization step, each side of the equation is evaluated only once given the current function, so no infinite regress arises. Ideally, the optimization converges to zero loss, thus finding mutually consistent update functions. Even if the loss does not converge to zero, we can still try to find update functions that are as consistent as possible.
>
> > I am concerned about the scalability of this approach.
>
> Regarding your third point, scalability is naturally a concern. However, we believe this can be addressed in future work and should not be a reason to reject the paper, as we evaluate on the most common games in the differentiable games literature and provide interesting theoretical results.
>
> Points 4 and 5 are great suggestions. We added these results and updated the paper (see Figure 1, Figure 7, Figure 10, Figure 12 and Figure 13).

---

> > ### Comment · Reviewer_hj7q · 2021-11-25
> > **Response to Rebuttal**
> >
> > I would like to thank the authors for their detailed responses to my concerns and for making the changes to the paper accordingly.
> > Please find my response to the rebuttal below.
> >
> > > Hence, it is unclear when COLA should be used instead of SOS.
> >
> > I agree that the increased consistency in COLA can result in more robustness to higher look-ahead rates. However, it is unclear to me why it is important and/or beneficial to support high look-ahead rates. The look-ahead rate is a tunable hyperparameter, and it makes sense why LOLA/SOS can diverge when an unreasonably high look-ahead rate (e.g., 10.0) is chosen. In practice, a reasonable look-ahead rate (e.g., 1.0) is chosen, and SOS shows stable performance with the theoretical guarantee. Hence, I partially agree with the benefit of COLA, and I am still concerned whether COLA should be preferable over SOS both practically and theoretically.
> >
> > > I am concerned about the scalability of this approach.
> >
> > One of the benchmark differential games in the literature is the Gaussian mixture domain motivated by the widely used GAN, and the generator and discriminator networks are deep neural networks with 6 layers (please refer to the SOS paper for details). Because the dimensions of $\theta_1$ and $\theta_2$ are large, the proposed method would have a difficult time solving this benchmark domain. While I agree that this can be a future direction, scalability is one major limitation of COLA which would be not easy to address within the proposed framework.
> >
> > > Could you clarify further on the consistency (Definition 1) regarding how it avoids the infinite regress problem?
> >
> > Thank you for the clarification.
> >
> > **Summary:** I believe this paper studies an important problem of MARL. However, I am still unsure about the benefit of COLA and agree with the concerns regarding writing (suggested by the other reviewers). Thus, I am leaning towards keeping my score.

---

> > > ### Author Response · Authors · 2021-11-26
> > > **Response 1: Response regarding benefits of COLA and scalability**
> > >
> > > Thank you for your comments on our response.
> > >
> > > > However, it is unclear to me why it is important and/or beneficial to support high look-ahead rates. The look-ahead rate is a tunable hyperparameter, and it makes sense why LOLA/SOS can diverge when an unreasonably high look-ahead rate (e.g., 10.0) is chosen. In practice, a reasonable look-ahead rate (e.g., 1.0) is chosen, and SOS shows stable performance with the theoretical guarantee.
> > >
> > > We agree that divergence at high look-ahead rates is not always a problem per se, as one can choose lower rates (note that the specific learning rate of 10.0 was just one example—we also demonstrated divergence in the Tandem game at a learning rate of 1.0, for instance, in Appendix I Figure 4b). However, we do believe that there are independent benefits to COLA’s greater robustness to higher look-ahead rates, such as faster convergence (in Proposition 5, we prove that in the Hamiltonian game, the convergence speed of consistent linear update functions strictly increases with increasing look-ahead rates) and requiring less tuning (at least when it comes to learning rates). Moreover, higher look-ahead rates often lead to more prosocial solutions. E.g., in the Ultimatum game, the solutions found by algorithms at a high look-ahead rate of 5.0 are fairer on average, but the losses achieved by LOLA and SOS have a high standard deviation, while COLA consistently finds a fair solution (Figure 10).
> > >
> > > > While I agree that this can be a future direction, scalability is one major limitation of COLA which would be not easy to address within the proposed framework.
> > >
> > > We want to add to our previous responses that predicting the gradient update or weights of a neural network is an active field of research (e.g. see Self-Tuning Networks https://arxiv.org/pdf/1903.03088.pdf and many others), which might allow our framework to benefit from any progress made in that field.
> > >
> > > Overall, we believe that given our edits to the paper, it would be valuable to the community, as it corrects claims made in the literature and presents a method with interesting theoretical and empirical results.
> > >
> > > Thank you very much for your time and your helpful comments, which allowed us to improve the paper and clarify our own thinking.

---

> > > > ### Author Response · Authors · 2021-11-27
> > > > **Conceptual appeal of COLA**
> > > >
> > > > To add to our previous response, we’d like to emphasize that we see one of COLA’s main benefits in its conceptual appeal. LOLA assumes that the other players are naive learners; this assumption is violated when two LOLA agents play against each other. SOS is an ad-hoc modification which adjusts LOLA to yield convergence guarantees. Both LOLA and SOS are useful, but it is natural to ask how a learning rule with consistent modelling assumptions would work, and what properties it might have.
> > > >
> > > > By introducing COLA, we make it possible to investigate this question. For instance, we show that COLA does not in general converge to stable fixed points, and thus inherits the “arrogance” that has been attributed to LOLA in prior work. We believe that such insights are valuable to the community, independently of COLA’s advantages over LOLA or SOS, e.g., in terms of better learning outcomes.

---

> > > > > ### Comment · Reviewer_hj7q · 2021-11-29
> > > > > **Response to Rebuttal**
> > > > >
> > > > > Thank you for the clarification. I have read the authors' latest responses, but they did not change my assessment. I agree with the conceptual benefit of COLA, and I believe that the paper can present this idea better in the future by improving multiple axes, such as stronger theoretical properties (not limited to a particular game (e.g., Tandem)) and better organization/writing of the paper (e.g., Section 6 can be further split into several sub-sections for improved clarity). Unfortunately, the current version of the paper seems to be not in the best form to be accepted.

---

> > > > > > ### Author Response · Authors · 2021-11-29
> > > > > > **Response to comment by reviewer hj7q**
> > > > > >
> > > > > > Thanks for your constructive feedback and for acknowledging the paper’s potential.
> > > > > >
> > > > > > > such as stronger theoretical properties (not limited to a particular game (e.g., Tandem))
> > > > > >
> > > > > > Regarding our use of particular games in proofs, we’d like to note that, e.g., our result in Proposition 4 shows that COLA does not in general recover SFPs (thus correcting the hypothesis that LOLA’s lack of guaranteed convergence to SFPs is due to its inconsistency); a single counterexample is sufficient to establish that general claim, as it shows that a general convergence guarantee is not possible. Similarly, to show that CGD’s series expansion does not in general recover higher-order LOLA (Proposition 2), or that COLA’s solutions are in general not unique (Proposition 3), it is sufficient to provide individual games as counterexamples.

---

> > > ### Author Response · Authors · 2021-11-29
> > > **Response 2**
> > >
> > > Dear Reviewer
> > >
> > > The discussion period will end today on the 29th of November. We would greatly appreciate if you could consider our latest response for a final review of our paper.
> > >
> > > Thank you again for having provided an insightful review and investing your time in this discussion.

---

> ### Author Response · Authors · 2021-11-24
> **Response 2**
>
> Dear Reviewer
>
> We hope we addressed your concerns about the empirical comparisons with our responses and the corresponding changes made to the paper. We ask you kindly to reply to our responses as soon as possible so that we still have enough time to discuss follow-up questions. The discussion period ends on the 29th of November.
>
> Thank you so much for your time.

---

### Author Response · Authors · 2021-11-15
**Joint Response**


Dear Reviewers

We appreciate your valuable feedback and want to thank you for helping us improve the paper. In this joint response, we address two major points that were common across reviewers (they will be addressed more context-specific in the direct responses).


### 1.) Proving that CGD is not equal to iLOLA:
We clarified the claim that the CGD paper makes (see “Contributions” in Section 1 and Section 4.2 in general). We also added a technically rigorous proof (see Proposition 2 and  Appendix D Proof of Proposition 2) to the paper.  The proof shows that exact CGD does not in general correspond to infinite-order LOLA (iLOLA). In particular, we show that the series expansion of CGD does not recover higher-order LOLA (HOLA), contrary to claims made in the CGD paper.  Our proof applies to both Taylor HOLA and exact HOLA.

### 2.) Advantages of COLA compared to SOS/LOLA?
A second major concern was the advantage of COLA over SOS/LOLA. First of all, COLA is conceptually preferable since LOLA and SOS are inconsistent in their assumption that the opponent is a naive learner.

Another significant benefit is COLA’s empirical robustness to a wider range of look-ahead rates. For example, in the Tandem game SOS and LOLA start diverging at a look-ahead rate of 0.5, whereas COLA still converges at look-ahead rates of 1 and higher (see Figure 4b). There is an inherent trade-off between too low look-ahead rates (no opponent shaping) and too high look-ahead rates (divergence for SOS/LOLA). Therefore robustness to higher look-ahead rates is attractive because we can introduce more opponent shaping without risking divergence. We also added a proof (see Appendix 3 Proof of Proposition 3) showing that, in contrast to LOLA and SOS, any linear solution to the consistency equations converges to the stable fixed point at the origin for any initial parameters and any look-ahead rate > 0 on the Hamiltonian game, further motivating the claim of robustness to look-ahead rates.

To further empirically compare COLA, SOS and, LOLA and motivate our claim to robustness, we also extended our results. We added a figure showing that SOS and LOLA diverge on the Tandem game for a look-ahead rate of 1.0 whereas COLA converges (see Figure 4b). On Matching Pennies at high look-ahead rates, SOS and LOLA mostly don’t converge whereas COLA converges even faster with a high look-ahead rate (see Figure 2b). For the Ultimatum game we added empirical results that show that COLA is the only method that finds the fair solution consistently at a high look ahead rate, whereas SOS and LOLA do not find it (see Figure 10d). In the Chicken game (newly added), we observe that SOS and Taylor LOLA crash whereas COLA swerves at high look-ahead rates (see Figure 12d). Crashing in the Chicken game results in a catastrophic payout for both agents and is generally to be avoided, whereas swerving results in a jointly preferable outcome.

We again thank reviewers for their feedback and believe that these new results have helped us to substantially improve the paper based upon it.

---

### Author Response · Authors · 2021-11-17
**Joint Response 2**

Dear Reviewers

In order to further support the claims made in our paper by theoretical results, we have added more proofs. First, we show that COLA's solutions are ***not*** necessarily unique (see Proposition 3 and Appendix E). Second, in addition to the experimental evidence, we now also prove that COLA does, in general, ***not*** recover SFPs (see Proposition 4 and Appendix F). Third, we provide an example (the Hamiltonian game) for which we prove that COLA converges under a wider range of learning rates than LOLA (see Proposition 5 and Appendix G).

Moreover, based on your feedback, we rewrote the Results section. First, we made sure not to overgeneralize any empirical observations. Second, we highlighted the advantages of COLA over other state-of-the-art general-sum algorithms such as LOLA, SOS, and CGD.

We want to thank you again for the informative feedback, which was very helpful in improving the paper.

---

### Decision · Program_Chairs · 2022-01-20

**Decision:**

Reject

**Comment:**

The main contribution of this paper is that it points out incorrect claims in the literature of multi-agent RL and provides new insight on the failure modes of current methods. Specifically, this paper investigates the inconsistency problem in LOLA (meaning it assumes the other agent as a naive learner, thus not converging to SFPs in some games). It then shows problems with two fixes in the literature: 1) HOLA addresses the inconsistency problem only when it converges; otherwise, HOLA does not resolve the issue. 2) GCD does not resolve the issue although it claims to do so. This paper then proposes a method COLA that fixes the inconsistency issue, which outperforms HOLA when it diverges. Reviewers generally agree that the insight from this work is interesting and important for the field. However, there were some concern on both the theory and the experiments. While the updated version addresses some of the concerns, it also made significant changes to both the theoretical and the empirical sections, and would benefit from another round of close review. Thus, I think the current version of this work is borderline.